# Fairness Artificial Intelligence in Clinical Decision Support: Mitigating Effect of Health Disparity

Hang Wu, Yuanda Zhu, Wenqi Shi, Li Tong, and May D Wang

*Georgia Institute of Technology*

Atlanta, USA

{hangwu,yzhu94,wqshi,ltong,maywang}@gatech.edu

*Abstract*—The United States, as well as the global community, experiences health disparities among socially disadvantaged populations. These disparities often manifest in the data utilized for AI model training. Without appropriate de-biasing strategies, models trained to optimize predictive performance may inadvertently capture and perpetuate these inherent biases. The utilization of biased models in clinical decision-making can inflict harm upon patients from disadvantaged groups and exacerbate disparities when these decisions are documented and employed to train subsequent AI models. Unlike conventional *correlation-based* methods, we aim to mitigate the negative impacts of health disparity by answering a *causal inference* question for fairness: *would the clinical decision support system make a different decision if the patient had a different sensitive attribute (e.g., race)?* Recognizing the high computational complexity of developing causal models, we propose a flexible and efficient causal-model-free algorithm, `CFReg`, which provides causal fairness for supervised machine learning models. In addition, `CFReg` also develops a novel evaluation metric to quantify fairness within clinical settings. We first validate `CFReg` using a healthcare dataset of 48,784 patients focused on care management, then generalize to another four benchmark datasets with racial and ethnic disparity, including law school admission, adult income, criminal recidivism, and violent crime prediction. Experimental results demonstrate that `CFReg` outperforms baseline approaches in both fairness and accuracy, achieving a good trade-off between model fairness and supervised classification performance.

## I. Introduction

Health disparity refers to the fact that certain disadvantaged social groups persistently experience worse healthcare treatment or higher health risks than advantaged social groups [1]. Such social groups include, but are not limited to, the poor, racial/ethnic minorities, and people from rural areas [2]. Health disparities in the US primarily refers to racial/ethnic disparities [1]. The racial and ethnic disparity encompasses a wide range of topics, including mortality, fatality, morbidity, health risks, and healthcare resource allocation. Multiple studies have shown the existence of racial disparities in health domains, including patients with cancers [3] and patients who have undergone cardiac surgery [4]. These disparities can be attributed to the fact that racial and ethnic minorities often receive lower-quality healthcare [5].

The disparity evident in reality is reflected in the big data we have acquired. When we utilize such data to train clinical decision support systems without considering these bias, the system will replicate the bias from the real-world data. For example, Obermeyer et al. [6] demonstrate that within a commercial software utilized to allocate patients to high-quality care programs, there exists a significant disparity in illness severity between black and white patients, as measured by the number of uncontrolled illnesses. The replication of bias in the learned models will in turn bring unfair impact on people of sensitive attributes.

In this study, we focus on developing fair supervised machine learning models to mitigate racial disparity in health decision support systems. There are mainly three types of fairness metrics in machine learning models: (1) **Group-based fairness metrics** measure differences in model behavior among subgroups with different sensitive attributes. For instance, they assess disparities in prediction accuracy between an advantaged group and a disadvantaged group [7]. (2) **Individual-based fairness metrics** emphasize the similarity in model decisions for similar individuals from opposite sensitive attribute groups [8]. However, the current group-based and individual-based metrics mainly rely on correlations and may evaluate (un)fairness based on spurious correlations [9]. Consequently, their reliability in practical applications is limited. As a result, (3) **Causality-based fairness metrics** [10], [9] aim to address the fundamental question: *Would the machine learning model[1] have made a different decision if the individual had a different sensitive attribute?*

Despite the intuitive definition, the practical application of these causality-based metrics and their corresponding learning algorithms is constrained in real-world scenarios [11], [12]. To begin, we leverage the outcome of a patient of a specific sensitive attribute (e.g., in the advantaged group) the *factual outcome*. For the same patient, if they had an opposite sensitive attribute (e.g., in the disadvantaged group), the outcome we would have observed is *counterfactual outcome*. To answer the fundamental question, we need to predict the counterfactual outcomes of a patient: their outcome if they were from the opposite sensitive feature group. An exact answer to the prediction of counterfactual outcomes requires understanding the causal relationships among all variables in our study. For each variable (e.g., blood pressure), we need to know what variables can cause it to change (e.g., stress level) and the magnitude of such changes. However, knowing all causal relationships is not possible in practice; causal inference-based fairness is

This research was supported by a Wallace H. Coulter Distinguished Faculty Fellowship (M. D. Wang), a Petit Institute Faculty Fellowship (M. D. Wang), and Microsoft Research.

---

[1]To avoid confusion, moving forward, we use "model" to exclusively denote the model describing causal relationships among all variables, and "hypothesis" to denote predictors such as regressors and classifiers.

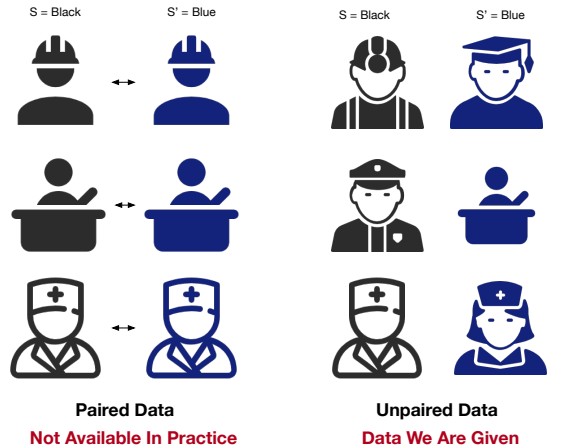

| S = Black | S' = Blue | S = Black | S' = Blue |

**Paired Data**
**Not Available In Practice**

**Unpaired Data**
**Data We Are Given**

Fig. 1: A comparison between paired dataset and unpaired data: in paired data, we have a one-to-one correspondence where, for an individual, we observe their outcome in both group $S$ and the opposite group $S'$.

generally very challenging to apply in real-world scenarios. In addition, even if we can approximate or learn a causal model as the ground truth, operating on it is also very challenging due to the large computational requirements. For example, some algorithms [10] require reasoning over distributions of unknown variables, or assumed structural equation models such as path-based counterfactual fairness [13].

To make counterfactual fairness more applicable in addressing health decision-making problems, we present CFReg, a flexible learning algorithm for approximating Counterfactual Fairness Regularization on predictive models without strong assumptions on the underlying causal knowledge. Instead of assuming or estimating a ground-truth causal model, we propose to directly learn mappings that infer what features an individual might have had if they had been in the opposite group of sensitive attributes.

If we had paired data between an individual and their counterfactual outcome, we would be able to build a supervised mapping to perform the transformation. In practice, however, the counterfactual nature restricts us from having access to the paired data from the two domains. For example, if we observe a black patient who was not assigned to a better health program, we would not be able to know what the model outcome would be if this patient were white while all other conditions remain the same (Fig. 1). Inspired by cycle generative adversarial networks (CycleGANs) [14], we propose learning such mappings using distribution matching techniques (Fig. 2). Based on these two mappings, we define our counterfactual fairness regularization term by minimizing the difference in prediction between an individual and their counterfactuals. The term is differentiable and thus can be combined with any gradient-based optimization algorithms, such as logistic regression and deep neural networks.

Our contributions are mainly three-fold:

- We propose a causal model-free method for measuring the counterfactual fairness of any machine learning models. By eliminating the requirement for causal structures, this metric is more efficient and generalizable in real-

world scenarios.
- Based on this metric, we develop a regularized learning algorithm, CFReg, to enforce counterfactual fairness on supervised learning models. It can be combined with any gradient-based optimization method. We also demonstrate how this metric effectively reduces the number of samples that would have been treated differently if they were assigned the opposite sensitive attribute.
- Experimental results on health and other datasets show the effectiveness of CFReg compared to multiple model-free baselines as well as model-based algorithms. Our algorithm naturally balances between accuracy and fairness, which is desirable in real-world cases with regulatory requirements.

## II. RELATED WORK

**Causal and Counterfactual Fairness.** Counterfactual fairness compares the decision of a model for an input sample to its counterfactual version [10]. To reason about counterfactual fairness, current methods mainly rely on a well-specified causal graph, either through expert knowledge in real-world scenarios or by assumption in simulation studies. For example, Kilbertus [9] studies counterfactual fairness metrics based on a structural causal model for the input data and imputes the counterfactual outcome of the decision model using Pearl's do-calculus for fairness comparisons. Extensions to model-based discussions include studying path-specific fairness [13] and developing a unified measurement for counterfactual fairness [15]. Based on counterfactual fairness metrics, researchers have developed algorithms for auditing the fairness of black-box models. For example, Black et al. use optimal transport algorithms to generate counterfactual input for testing the fairness of machine learning models [16]. In addition, Ustun et al. [17] discuss how an individual can improve their outcome by changing their attributes in a linear classification setting. In contrast, our work focuses on developing a counterfactual fairness model when a ground-truth causal model is not available. Moreover, our work can also be applied to regression settings, which are not covered by most existing work.

**Distribution Matching.** Relating two domains and mapping samples from one domain to another has wide applications in machine learning and data analytics, for example, image-to-image translation [18] and language translation [19]. The problem becomes more manageable when dealing with paired samples. For example, in a language translation context, during the training phase, we possess a sentence in a source language along with its corresponding translation in the target language. This scenario can be formulated as a standard supervised learning problem by employing techniques such as transformers [20]. Learning from unpaired data poses a greater challenge compared to paired data due to the limited supervision available in our setting. CycleGAN [14] introduces a cycle-consistency loss, which ensures that the learned mappings are reversible. This inspires us to map a sample from one domain to another and then map it back without any loss. In our work, we leverage CycleGAN to generate counterfactual outcomes, essentially learning mappings between the factual domain and the counterfactual domain.

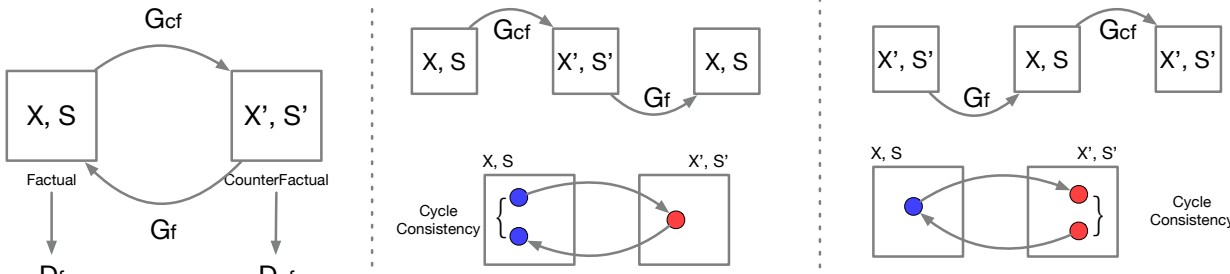

Fig. 2: An overview of the proposed approach: We apply cycleGANs to learn mappings from a person with one sensitive attribute to another. When training a supervised learning model, for a person $(x, s)$, we apply the mappings to obtain $(x', s')$ and require that the model's prediction is similar to $(x, s)$ and $(x', s')$.

## III. METHOD

### A. Problem Formulation: Counterfactual Fairness

We study supervised learning and its fairness; for presentation simplicity, we utilize binary classification as an illustrative example. However, the discussion can be extrapolated to other scenarios, including regression and multi-class classification. For a binary classification problem, our objective is to learn a hypothesis $h : (x, s) \to y \in \{0, 1\}$, where $x$ represents the feature of a sample, $s$ denotes the sensitive attribute, and $y$ corresponds to its label. The counterfactual fairness of the classification hypothesis $h$ is defined as follows:

**Definition 1.** A hypothesis $\hat{y} = h(x)$ is counterfactually fair if $\forall x, s$,

$$Pr(\hat{y}|X = x, S = s) = Pr(\hat{y}_{do(s)=s'}|X = x, S = s). \quad (1)$$

We adopt the notation of Pearl's do-calculus. For a sample of $(x, s)$, $do(s) = s'$ denotes flipping the sensitive attribute from $s$ to $s'$ (e.g., changing gender from male to female) and observing the corresponding changes in the children variables of $s'$ that change $x$ to $x'$. We require that the model outcome remains unchanged for an individual in group $s$ if they were in another group $s'$.

We note that this is an individual-level fairness concept, which is a much stronger notion than group-based fairness metrics. Enforcing individual-level fairness leads to an improvement in fairness overall. To understand counterfactual fairness as an individual fairness metric, we define a pair $((x_1, s_1), (x_2, s_2))$ to be similar if $(s_1 \neq s_2)$ and $(x_1 = x_2|do(s_2 \to s_1))$. The decision is considered similar if $Pr(\hat{Y}_1) = Pr(\hat{Y}_2)$, ensuring that similar individuals receive similar treatment.

Furthermore, our model $h$ takes the input of both the features and sensitive attributes $x, s$. In practice, the sensitive attribute $s$ may or may not be used. When $s$ is not used as a model input, this corresponds to fairness through the unawareness algorithm [8]. However, removing $s$ from the model input does not necessarily help mitigate the unfairness of the machine learning algorithm, as there can be correlated variables of $s$ in $x$. We discuss the difference between the two versions in our experiment sections.

### B. Learning Counterfactual Mappings

To identify the counterfactual outcome of an individual, $g : (x, s) \to (x', s')$, we approach it as the learning of a pair of mappings between two domains, $S_0 = \{(x, s = 0)\}$ and $S_1 = \{(x, s = 1)\}$. We train $g_{S_0 \to S_1}$ to map a sample from $S_0$ to $S_1$, and another $g_{S_1 \to S_0}$ to map in the opposite direction.

Given the counterfactual nature of the real world, we fail to observe the counterfactual outcome, $(x', s')$ for an individual $(x, s)$. Consequently, we lack a paired dataset for directly learning the mappings. Instead, we are provided with an unpaired dataset, $D_0 = \{(x_i, s_i = 0)\}$, sampled from the domain $S_0$ and $D_1 = \{(x_i, s_i = 1)\}$ from $S_1$. Our goal is to identify the mappings that transform samples from one domain to the other. To achieve this, we adopt the workflow of CycleGAN [21] that augments conventional GAN training with a cycle consistency loss.

Let us briefly revisit CycleGAN and use learning $g_{S_0 \to S_1}$ as an example. Alongside $g_{S_0 \to S_1}$, we set up a discriminator, $disc$, to classify samples from $S_0$ and $S_1$. We assume that $g_{S_1 \to S_0}$ is well trained. Consequently, the overall objective for $g_{S_0 \to S_1}$ consists of two parts, including:

- Adversarial loss $L_{GAN}$:

$$\begin{aligned} L_{GAN} = \ & \mathbb{E}_{(x,s=0)\sim S_0}[\log disc(x, 0)] \\ & + \mathbb{E}_{(x,s=1)\sim S_1}[\log disc(g_{S_0 \to S_1}(x, 1))] \quad (2) \end{aligned}$$

- Cycle-consistency loss $L_{Cyc}$:

$$\begin{aligned} L_{Cyc} = \ & \mathbb{E}_{(x,s=0)\sim S_0}[\|g_{S_1 \to S_0}(g_{S_0 \to S_1}(x, 0)) - (x, 0)\|_1] \\ & + \mathbb{E}_{(x,s=1)\sim S_1}[\|g_{S_0 \to S_1}(g_{S_1 \to S_0}(x, 1)) - (x, 1)\|_1] \quad (3) \end{aligned}$$

During training, we iteratively update $g_{S_0 \to S_1}$ and $g_{S_1 \to S_0}$ until the algorithm converges. The advantage of neural network-based mappings lies in their flexibility; theoretically, neural networks can accurately fit any nonlinear function, providing us with significantly higher modeling capabilities compared to mappings defined by optimal transport [21].

### C. Regularization via Counterfactual Mappings

With two mappings, $g_{S_0 \to S_1}$ and $g_{S_1 \to S_0}$, we can construct our proposed counterfactual regularization as follows. For a

**Algorithm 1:** Counterfactual Fairness

**Data:** $D_0, D_1$

Fit CycleGAN on $D_0, D_1$ to learn two mappings
$g_{S_0 \rightarrow S_1}, g_{S_1 \rightarrow S_0}$ ;

Initialize classifier as $h : (x, s) \rightarrow [0, 1]$ ;

**while** *Not Converged* **do**

  Sample a batch of data $B = \{(x, s, y)\}$ from both
$D_0, D_1$ ;

  Compute predicted outcome as $\hat{y} = h(x, s)$ ;

  Construct classification loss $L_{clf}$ by comparing the
prediction $\hat{y}$ to the ground truth $y$ ;

  Build counterfactual samples as $(x', s')$ for each
$(x, s)$ using learned mappings $g_{S_0 \rightarrow S_1}, g_{S_1 \rightarrow S_0}$ ;

  Compute counterfactual outcome $y' = h(x', s')$,
and construct counterfactual loss as
$L_{cf} = D(y, y')$ ;

  Back propagate total loss as $L = L_{clf} + \lambda L_{cf}$ and
update $h$ parameters ;

**end**

**Result:** h

---

sample consisting of $(x, s)$, we can acquire:

$$
\begin{aligned}
dist(Pr(\hat{Y} = 1 | X = x, S = s), \\
Pr(\hat{Y} = 1_{do(s) = s'} | X = x, S = s)) \\
= dist(f(x, s), f(g_{s \rightarrow s'}(x), s')), \quad (4)
\end{aligned}
$$

where $dist(a, b)$ is a distributional divergence function. For instance, consider the KL divergence, where $dist(a, b) \geq 0$ and $D(a, b) = 0$ if and only if $a = b$. Therefore, minimizing the divergence is tantamount to minimizing the metrics of counterfactual fairness. Since this objective is differentiable, we can incorporate this regularization similar to conventional L2 regularization.

### D. Overall Algorithm

The overview of CFReg is in **Algorithm 1**. The computed loss $L_{cf}$ serves a dual purpose: it acts as a regularization term during training and can also be used to evaluate the counterfactual fairness of an already trained classifier $h$. The hyperparameter $\lambda$ controls the trade-off between accuracy and fairness. In the experimental section, we further demonstrate our model's sensitivity to different choices of $\lambda$.

### IV. THEORETICAL ANALYSIS

In this section, we analyze the theoretical aspects of our algorithm, focusing on how it effectively reduces the unfairness of learned supervised learning algorithms.

### A. FlipSets and Regularization

We use the *flipset* in Black et al. [16] as a proxy for analyzing the counterfactual fairness of a binary classifier $h$.

**Definition 2.** Let $h : (x, s) \rightarrow \{0, 1\}$ be a classifier and $g_{0 \rightarrow}$ be the mapping from $S_0 : \{x, s = 0\}$ to $S_1 : \{x, s = 1\}$. The flipset $Flip(h, g_{S_0 \rightarrow S_1})$ is the set of points that changes its label after the counterfactual mapping $x' = g_{S_0 \rightarrow S_1}(x)$,

$$
Flip(h, g_{S_0 \rightarrow S_1}) = \{x \in S_0 | h(x, 0) \neq h(x', 1)\}. \quad (5)
$$

We can similarly define $Flip(h, g_{S_1 \rightarrow S_0})$ for the reverse case, and $Flip(h)$ as the union of the $Flip(h, g_{S_0 \rightarrow S_1}), Flip(h, g_{S_1 \rightarrow S_0})$. The smaller the size of flipset, the more fair a model $h$ is. Note here the slight difference in our definition is that we used the cycleGAN instead of the optimal transport mapping. We then show that enforcing counterfactual fairness metrics strictly upper bounds the size of flipsets.

**Proposition 1.** For a classifier $h$ with a counterfactual fairness metric of $\epsilon$, where the distribution divergence uses KL divergence, the size of flipsets denotes as $|Flip(h)| \leq 2\sqrt{2}\epsilon^2 * N$, where $N$ is the size of the dataset.

*Proof:* Let $P, Q$ denote the distribution of $\hat{Y} | X = x, S = s$, and $\hat{Y} | X = x', S = s'$, for $(x, s), (x', s')$ are the pair of observed data and its counterpart. Using Pinsker inequality:

$$
\|P - Q\| \leq \sqrt{2KL(P||Q)} = \sqrt{2\epsilon}, \quad (6)
$$

where $\|P - Q\|$ is the total variational norm distance between $P, Q$. Specifically, for two Bernoulli variables,

$$
\|P - Q\| = \frac{1}{2} \sum_{\hat{Y} = 0, 1} |P - Q|, \quad (7)
$$

where $P - Q$ is the probability difference when the factual outcome and counterfactual outcome differ. Thus, converting probability to counts, we can obtain:

$$
|Flip(h)| \leq 2\sqrt{2}\epsilon^2 * N. \quad (8)
$$

### B. Counterfactual Fairness and Demographic Parity

Built upon the Proposition 2 from [16], we can further use our proposed technique to enforce demographic parity and equalized opportunity.

**Proposition 2.** For a classifier $h$, and an exact mapping of $g_{S_0 \rightarrow S_1}$ and $g_{S_1 \rightarrow S_0}$, if the counterfactual fairness metric is 0, the model also satisfies demographic parity.

*Proof:* As Prop. 1 shows, when the counterfactual fairness metric is 0, the $|Flip(h)| = 0$. Thus, plugging in the proposition from Black et al., the demographic parity is satisfied:

$$
|\{x \in S_0 | h(x, 0) = 1\}| = |\{x' \in S_1 | h(x', 1) = 1\}|. \quad (9)
$$

### V. RESULTS

#### A. Datasets

For a comprehensive evaluation, we examine CFReg on five different datasets related to health and social fairness:

*1) Law Data:* This dataset contains information on 21,790 law school students. We build supervised learning models to predict (normalized) first-year grades based on five features: undergraduate GPA, LSAT scores, race, and sex. We convert this to a classification problem by assigning a label $Y = 1$ if a student's GPA is above the median. The sensitive attribute is race (white or non-white).

*2) Health Data:* We use a synthetic dataset [6], which closely approximates the original data distribution while adhering to privacy regulations. It contains 48,784 observations and 160 features, including demographic information, comorbidity

data, and biomarker/medication information. Given the 186 variables and lack of a ground-truth causal model, we only compare results without relying on a causal model, reflecting more prevalent real-world scenarios.

*3) Chicago Strategic Subject List (SSL):* This dataset[2] aims to identify individuals likely to be involved in violent crimes, either as victims or perpetrators. We use eight features to calculate a risk score (0-500) for each individual's likelihood of being involved in a shooting, which we then threshold into a binary label. The sensitive attribute is race, with black subjects more likely to receive a positive label in our experiments due to different thresholds for black and white subjects.

*4) ProPublica COMPAS:* The ProPublica COMPAS (Correctional Offender Management Profiling for Alternative Sanctions) dataset[3] contains recidivism likelihood scores for defendants. ProPublica's original analysis identified discriminatory outputs, with higher likelihoods assigned to black subjects.

*5) Adult Income:* Extracted from the 1994 Census dataset[4], this dataset predicts whether an individual's annual income exceeds \$50,000. We use sex as the sensitive attribute and preprocess the data using the IBM toolkit [22].

### B. Baselines

*1) Unregularized Classifier (LR):* This is the vanilla classification algorithm without fairness constraints. For fair comparison, all baselines including `CFReg` also use logistic regression as classifier $h$, with the only difference being the fairness regularization technique applied.

*2) Causal Model-based Counterfactual Fairness:* Kusner et al. [10] propose three levels of causal model-based learning algorithms:

- Level 1 (L1): Builds $\hat{Y}$ using only the non-descendants of the sensitive attributes.
- Level 2 (L2): Performs MCMC sampling on the causal model's hidden variables and predicts $\hat{Y}$ using the inferred distribution of hidden variables.
- Level 3 (L3): Assumes a deterministic linear model with hidden variables, calculates them from the data, then uses these calculated hidden variables for constructing $\hat{Y}$.

*3) k-Nearest Neighbor Matching (k-NN):* It originates from the matching algorithm in treatment effect estimation [23]. For $g_{S_0 \to S_1}$, we construct:

$$g_{S_0 \to S_1}(x, s = 0) = \arg\min_{x' \in S_1} \|x' - x\|_2. \quad (10)$$

We similarly construct $g_{S_1 \to S_0}$. The rest of the training algorithm remains the same as in Alg. 1, except for Line 1.

*4) Optimal Transport (OT) Matching:* OT uses optimal transport to build the mappings $g_{S_0 \to S_1}$ and $g_{S_1 \to S_0}$. While there are many variants of OT, such as Sinkhorn iteration, entropic regularization, kernel OT, and linear OT, our experiments show that linear OT obtains the best performance. Other algorithms face convergence issues or perform worse than simple linear OT. We use the implementation from the POT library [24].

[2] https://data.cityofchicago.org/Public-Safety/
[3] https://www.propublica.org/
[4] http://archive.ics.uci.edu/ml/datasets/adult

TABLE I: Benchmark comparison on law dataset with sensitive attributes race. $-s$ indicates that the hypothesis $h$ includes sensitive attributes as part of the input. CF denotes `CFReg`. Notations are consistent across tables.

| Alg | Counterfactual Fairness | | | | Group-Based Fairness | | Classification | | | Trade-offs |
|---|---|---|---|---|---|---|---|---|---|---|
| | $L_{CF}$ | Flip | Flip0 | Flip1 | DemPar | $\Delta TPR$ | F1 | AUC | Acc | ratio |
| LR | 1.1 | 48.2 | 40.8 | 7.4 | 0.281 | 0.170 | 0.565 | 0.665 | 0.604 | NaN |
| LR-s | 1.0 | 45.4 | 39.2 | 6.2 | 0.412 | 0.130 | 0.596 | 0.665 | 0.611 | NaN |
| L1 | 113.2 | 1245.9 | 1054.9 | 191.0 | 0.280 | 0.196 | 0.599 | 0.637 | 0.597 | NaN |
| L2 | 0.645 | 66.8 | 53.4 | 13.4 | 0.026 | 0.272 | 0.559 | 0.576 | 0.559 | NaN |
| L3 | 6.6 | 136.9 | 115.3 | 21.6 | 0.013 | 0.260 | 0.574 | 0.607 | 0.580 | NaN |
| kNN | 1.2 | 48.1 | 42.8 | 5.3 | 0.268 | 0.159 | 0.559 | 0.665 | 0.605 | 5e2 |
| kNN-s | 1.1 | 42.5 | 35.6 | 6.9 | 0.430 | 0.119 | 0.594 | 0.665 | 0.609 | 1e3 |
| OT | 0.764 | 47.5 | 41.6 | 5.9 | 0.206 | 0.201 | 0.497 | 0.663 | 0.587 | 2e2 |
| OT-s | 0.755 | 39.6 | 36.3 | 3.3 | 0.390 | 0.111 | 0.554 | 0.663 | 0.599 | 2e2 |
| **CF** | **0.078** | **31.4** | **24.8** | **6.6** | **0.143** | **0.175** | **0.368** | **0.665** | **0.562** | **2e5** |
| **CF-s** | **0.239** | **36.6** | **29.7** | **6.9** | **0.416** | **0.122** | **0.596** | **0.665** | **0.612** | **4e3** |

### C. Evaluation Metrics

We report the following metrics for a comprehensive evaluation on both classification and fairness. We leverage accuracy, AUC, and F1 for classification performance. For counterfactual fairness, we report $|Flip(h)|, |Flip(h, S_0)|, |Flip(h, S_1)|$, short as Flip, Flip0, Flip1, and the CF loss $L_{CF}$. For group-based fairness, we report demographic parity ($DemPar = |P(\hat{Y} = 1|S = 1) - P(\hat{Y} = 1|S = 0)|$) and difference in true positive rates ($\Delta TPR = |P(\hat{Y} = 1|S = 1, Y = 1) - P(\hat{Y} = 1|S = 0, Y = 1)|$).

### D. Implementation Details

We train each classifier for a maximum of 10 epochs with early stopping, splitting the data $4:1$ for training and testing. We use the Adam optimizer with a learning rate of $0.0003$. Each algorithm is run with 5-fold cross-validation, and we compute the mean and standard deviation for each metric. For each algorithm, we vary $\lambda$ and choose the best value based on the trade-offs between accuracy and fairness. We compare each algorithm $A$ to the unconstrained algorithm $LR$. If algorithm $A$ incurs a loss in accuracy while increasing the fairness metric, we compute the ratio: $\frac{|L_{CF}(A) - L_{CF}(LR)|}{|Acc(A) - Acc(LR)|}$, which serves as a proxy for evaluating the trade-off between accuracy and fairness. A higher ratio indicates that the algorithm achieves a better trade-off.

### E. Results on Law Data

**Fairness.** On fairness, we first focus on the counterfactual loss metric $L_{CF}$. As shown in Table I, CF-s and CF have the lowest counterfactual losses compared to other baselines (OT, kNN, and causal model-based L1, L2, and L3). Between CF and CF-s, we observe that CF further reduces $L_{CF}$ at the cost of reduced classification performance metrics (F1 score drops from 0.596 to 0.368, recall drops from 0.580 to 0.256). Additionally, our algorithms (CF, CF-s) also achieve the best fairness performance in terms of flipset size (Flip, Flip0, Flip1). For group-based fairness, causal model-based baselines (L2 and L3) have the lowest demographic parity scores, while our algorithms (CF, CF-s) do not show lower differences in true positive rates ($\Delta TPR$).

TABLE II: Benchmark comparison on health dataset with sensitive attributes race.

| Alg | Counterfactual Fairness | | | | Group-Based Fairness | | Classification | | | Trade-offs |
|---|---|---|---|---|---|---|---|---|---|---|
| | $L_{CF}$ | Flip | Flip0 | Flip1 | DemPar | $\Delta TPR$ | F1 | AUC | Acc | ratio |
| LR | 2311.0 | 2023.2 | 239.9 | 1783.2 | 0.281 | 0.170 | 0.786 | 0.865 | 0.792 | NaN |
| LR-s | 2445.5 | 2088.3 | 248.0 | 1840.3 | 0.412 | 0.130 | 0.788 | 0.863 | 0.787 | NaN |
| kNN | 1989.2 | 1803.9 | 217.5 | 1586.4 | 0.268 | 0.159 | 0.792 | 0.873 | 0.791 | 4e5 |
| kNN-s | 2052.5 | 1582.3 | 197.1 | 1385.2 | 0.430 | 0.119 | 0.786 | 0.864 | 0.783 | 3e6 |
| OT | 2008.6 | 1877.2 | 228.2 | 1649.0 | 0.206 | 0.201 | 0.798 | 0.868 | 0.794 | 8e5 |
| OT-s | 2136.4 | 2025.9 | 227.1 | 1798.8 | 0.390 | 0.111 | 0.795 | 0.870 | 0.795 | 3e5 |
| **CF** | 1492.8 | 1931.8 | 317.3 | 1614.5 | 0.143 | 0.175 | 0.795 | 0.862 | 0.789 | 4e6 |
| **CF-s** | 1618.2 | 1801.9 | 308.0 | 1493.9 | 0.416 | 0.122 | 0.793 | 0.866 | 0.786 | 5e6 |

TABLE III: Benchmark comparison on Chicago-SSL dataset with sensitive attributes race.

| Alg | Counterfactual Fairness | | | | Group-Based Fairness | | Classification | | | Trade-offs |
|---|---|---|---|---|---|---|---|---|---|---|
| | $L_{CF}$ | Flip | Flip0 | Flip1 | DemPar | $\Delta TPR$ | F1 | AUC | Acc | ratio |
| LR | 4.3 | 212.9 | 49.9 | 162.9 | 0.055 | 0.643 | 0.469 | 0.864 | 0.846 | N/A |
| LR-s | 6.6 | 205.2 | 47.0 | 158.2 | 0.117 | 0.111 | 0.464 | 0.856 | 0.792 | N/A |
| kNN | 5.0 | 342.3 | 129.0 | 213.3 | 0.055 | 0.621 | 0.468 | 0.865 | 0.850 | 1570.4 |
| kNN-s | 7.4 | 232.0 | 52.7 | 179.3 | 0.118 | 0.113 | 0.465 | 0.856 | 0.793 | 16862.1 |
| OT | 2.1 | 138.2 | 24.4 | 113.8 | 0.047 | 0.655 | 0.458 | 0.864 | 0.846 | 92965.8 |
| OT-s | 0.632 | 61.9 | 31.1 | 30.8 | 0.149 | 0.026 | 0.473 | 0.857 | 0.813 | 11080.2 |
| **CF** | 2.0 | 132.3 | 16.3 | 116.0 | 0.042 | 0.596 | 0.452 | 0.864 | 0.855 | 96390.0 |
| **CF-s** | 0.457 | 29.6 | 13.0 | 16.6 | 0.125 | 0.001 | 0.470 | 0.856 | 0.828 | 81027.1 |

TABLE IV: Benchmark comparison on COMPAS dataset with sensitive attributes race.

| Alg | Counterfactual Fairness | | | | Group-Based Fairness | | Classification | | | Trade-offs |
|---|---|---|---|---|---|---|---|---|---|---|
| | $L_{CF}$ | Flip | Flip0 | Flip1 | DemPar | $\Delta TPR$ | F1 | AUC | Acc | ratio |
| LR | 7.8 | 57.0 | 25.9 | 31.1 | 0.155 | 0.065 | 0.555 | 0.733 | 0.669 | N/A |
| LR-s | 7.9 | 56.1 | 25.5 | 30.6 | 0.162 | 0.060 | 0.557 | 0.733 | 0.670 | N/A |
| kNN | 6.6 | 54.3 | 24.7 | 29.6 | 0.141 | 0.052 | 0.544 | 0.735 | 0.665 | 635.2 |
| kNN-s | 6.8 | 53.7 | 24.2 | 29.5 | 0.167 | 0.058 | 0.554 | 0.733 | 0.666 | 175945.8 |
| OT | 3.0 | 53.9 | 27.3 | 26.6 | 0.121 | 0.044 | 0.470 | 0.730 | 0.649 | 1577.1 |
| OT-s | 2.6 | 52.0 | 31.5 | 20.5 | 0.162 | 0.076 | 0.495 | 0.729 | 0.651 | 1457.3 |
| **CF** | 5.0 | 48.1 | 25.1 | 23.0 | 0.149 | 0.060 | 0.557 | 0.733 | 0.670 | 7147.9 |
| **CF-s** | 0.542 | 21.2 | 15.1 | 6.1 | 0.215 | 0.036 | 0.585 | 0.736 | 0.677 | 2522.0 |

**Classification.** For classification performance, we primarily focus on AUC and F1 metrics. We observe that causal model-based algorithms (L1, L2, and L3) achieve much lower AUC performance than other unregularized and regularized algorithms, with no clear difference among the latter. For F1 scores, algorithms using sensitive attributes consistently outperform their counterparts that do not use sensitive attributes (LR-s vs LR, kNN-s vs kNN, OT-s vs OT, and CF-s vs CF). The best-performing fair algorithm, CF-s, has a similar F1 performance to L1 and significantly outperforms L2 and L3. Overall, CF-s achieves the best classification accuracy when considering both F1 and AUC metrics.

**Trade-offs.** Our algorithm demonstrates a good trade-off between accuracy and fairness. The ratio of reduction in fairness over reduction in accuracy shows that our proposed algorithms (CF and CF-s) have the highest ratios. This confirms the effectiveness of our approaches: even without a ground-truth causal model, we can effectively reduce model unfairness by enforcing our proposed counterfactual fairness regularization techniques. Among the three levels of causal model-based algorithms, Level 1 cannot reasonably reduce unfairness while maintaining accuracy similar to the LR model, while Levels 2 and 3 reduce unfairness at a significant cost to accuracy.

### F. Results on Health Data

As shown in Table II, our algorithms CF and CF-s achieve the lowest counterfactual losses ($L_{CF}$), while maintaining flip set sizes similar to those of the kNN and kNN-s baseline models. In terms of group-based fairness, CF demonstrates the lowest demographic parity score. However, neither of our models show lower differences in true positive rates ($\Delta TPR$). All algorithms achieve performance comparable to the unregularized logistic regression models. This demonstrates our ability to trade off a slight reduction in classification performance for a substantial improvement in fairness. For instance, our CF algorithm improves the fairness metric by approximately 30% while only reducing AUC by 0.3%.

### G. Additional Experimental Results

**Chicago SSL.** As shown in Table III, our proposed approach CF-s demonstrates significantly lower counterfactual loss $L_{CF}$ and flipset sizes compared to all other models. Regarding group-based fairness, models without sensitive attributes generally show lower disparity scores. Notably, CF-s achieves a substantially lower difference in true positive rates ($\Delta TPR$) than all other models. In terms of classification performance, both CF and CF-s maintain performance comparable to the baseline models, similar to the healthcare dataset results.

**ProPublica COMPAS.** Table IV illustrates that our proposed approach CF-s achieves markedly lower counterfactual loss $L_{CF}$ and flipset sizes compared to all other models. In terms of group-based fairness, our proposed approaches (CF and CF-s) remain similar scores compared to other baseline models.

**Adult Income.** As evident from Table V, our proposed approach CF-s exhibits lower counterfactual loss $L_{CF}$ compared to other models. However, all models except the unregularized logistic regression models show similar flipset sizes. For group-based fairness, our proposed approaches (CF and CF-s) do not demonstrate significantly lower scores than other baseline models.

### H. Discussions

In this section, we discuss the trade-offs between accuracy and fairness and the parameter sensitivity of our approach. We first note that among the two types of fairness metrics, namely counterfactual fairness and group-based fairness, we need to choose which metrics to use in practice. For example, local laws and regulations may have specific requirements for algorithm fairness metrics. We advocate for counterfactual-based metrics because they are defined from an individual standpoint, allowing us to provide a fairness examination of model prediction results for each person. Additionally, we can aggregate these results into a scalar statistic, making them comparable to group-based metrics. Although the results are not normalized to the $[0, 1]$ range, in practice, we can always linearly scale the results for easier comparison. For example, the flipset metrics have a minimum of 0 and a maximum of the total sample size, so they could be normalized. We can also select the $L_{CF}$ of a baseline algorithm (for example, the Logistic regression algorithm) as the maximum, and normalize the metrics of other algorithms accordingly.

TABLE V: Benchmark comparison on Income dataset with sensitive attributes sex.

| | Counterfactual Fairness | | | | Group-Based Fairness | | Classification | | | Trade-offs |
|---|---|---|---|---|---|---|---|---|---|---|
| Alg | $L_{CF}$ | Flip | Flip0 | Flip1 | DemPar | $\Delta TPR$ | F1 | AUC | Acc | ratio |
| LR | 633.4 | 495.9 | 93.2 | 402.7 | 0.073 | 0.088 | 0.404 | 0.848 | 0.796 | N/A |
| LR-s | 679.8 | 643.7 | 89.4 | 554.3 | 0.079 | 0.088 | 0.397 | 0.847 | 0.797 | N/A |
| kNN | 505.5 | 409.9 | 84.4 | 325.5 | 0.068 | 0.067 | 0.372 | 0.845 | 0.795 | 39174.3 |
| kNN-s | 526.5 | 418.0 | 90.6 | 327.4 | 0.072 | 0.077 | 0.401 | 0.846 | 0.799 | 613057.0 |
| OT | 533.5 | 411.1 | 108.5 | 302.6 | 0.050 | 0.100 | 0.385 | 0.839 | 0.798 | 10961.5 |
| OT-s | 583.9 | 402.9 | 99.7 | 303.2 | 0.077 | 0.085 | 0.407 | 0.844 | 0.795 | 37113.2 |
| CF | 525.5 | 413.9 | 98.9 | 315.0 | 0.069 | 0.094 | 0.393 | 0.846 | 0.798 | 57969.5 |
| CF-s | 453.3 | 435.0 | 92.0 | 343.0 | 0.070 | 0.068 | 0.385 | 0.846 | 0.797 | 357959.3 |

To analyze the how choosing an appropriate $\lambda$ can effectively change the trade-off between fairness and metrics, we present how metrics change with the hyper-parameter $\lambda$ in Fig. 3. Our method has a better $L_{CF}$ consistently over all three datasets with different selections of $\lambda$. However, from the recall metric in Fig. 3 (c), as we increase $\lambda$, the recall quickly goes to zero, i.e., the algorithms tend to predict all instances to 0 to obtain a fairer predictions. This indicates the need in practice to have an overall view of all the classification and fairness metrics, and to select the best trade-offs.

For each algorithm, we compare two versions: one that uses sensitive attributes in the classifier, and one without (fairness through unawareness). We observe that for baseline logistic regression, using sensitive attributes increases the unfairness of the algorithm in terms of both $L_{CF}$ and demographic parity $DPar$. Conversely, for OT and CF, using sensitive attributes decreases the unfairness. We hypothesize these algorithms can exploit the hidden links between a sensitive attribute variable and its proxy variables, so adding sensitive attributes into the algorithm helps mitigate the unfairness of learned classifiers. Additionally, we note that $L_{CF}$ and $DPar, \Delta TPR$ may not necessarily correlate with each other when $\lambda$ is small.

## VI. CONCLUSION

Our causal model-free approach effectively reduces unfairness in learned supervised learning systems without sacrificing accuracy. Compared with baseline causal model-free algorithms and even causal model-based algorithms, `CFReg` shows better performance in both fairness and classification, as well as improved trade-offs in reducing unfairness at the cost of reduced prediction accuracy. We also demonstrate how the proposed regularization theoretically regularizes the counterfactual fairness metric. By building fair machine learning systems for clinical decision support applications, we may mitigate health disparities and improve health outcomes for patients from disadvantaged groups.

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

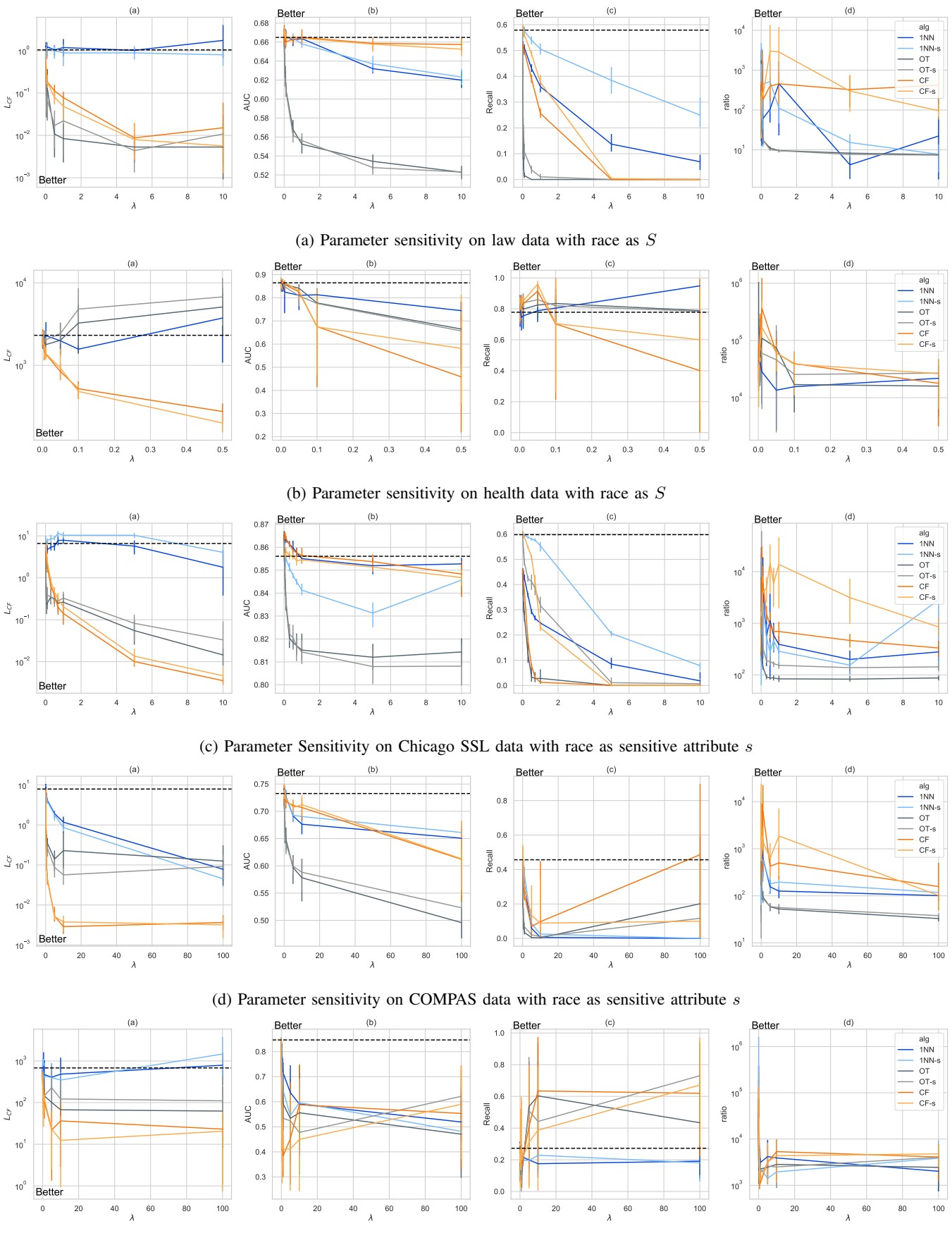

(a) Parameter sensitivity on law data with race as $S$

(b) Parameter sensitivity on health data with race as $S$

(c) Parameter Sensitivity on Chicago SSL data with race as sensitive attribute $s$

(d) Parameter sensitivity on COMPAS data with race as sensitive attribute $s$

(e) Parameter sensitivity on adult income data with sex as sensitive attribute $s$

Fig. 3: Parameter Sensitivity Results for $\lambda$: We find our algorithm, compared to baselines, has the best trade-offs between accuracy and fairness. In addition, when $\lambda$ is too high, all the algorithm tends to predict 0 for all instances, reaching a perfectly fair but zero useful model.