# OpenReview forum: "Fairness Artificial Intelligence in Clinical Decision Support: Mitigating Effect of Health Disparity"
_IEEE.org/EMBS/BHI/2024/Conference — IEEE BHI'24_

### Official Review · Reviewer_J9od · 2024-08-08
**A general purpose machine learning algorithm to improve fairness in AI models**

**Overall Rating:** 6
**Confidence:** 4

**Other Quality Metrics:**

(a) Clarity of writing; The paper is well-written, well-structured, and coherent. Small typos are suggested to fix.

(b) Clinical Significance; I believe that more clinically relevant dataset are needed in a future study to judge how significant the proposed algorithm can reduce the bias in machine learning results.

 (c) Methodological Novelty; It seems that the proposed CFReg method is novel with some derivation and insights from the already existing conventional methods.

 (d) Experiments and Results; the superiority of the proposed method has been shown in details and on 5 different dataset. Some improvements are suggested to better visualize the results and make the main part of the paper more concise.

**Questions For The Authors:**

The rationale and message of Fig. 1 might not be directly clear. Please better explain what one-on-one correspondance mean. Does it mean exactly the same person but with a different skin color?

In section II, it has been mentioned that ‘Moreover, our work can also be applied to regression settings, which are not covered by most existing work’. Please discuss whether your proposed model can be used to improve a regression-based motion artifact method introduced in https://doi.org/10.1109/tbme.2024.3438375, when information such as ethnicity or race of the test subjects are provided.

The three parts in Fig. 2 might be labeled with (a), (b), and (c) and better explained in the caption.

Section V.B mentions ‘our experiments show that linear OT obtains the best performance.’ while no reference is given to any Table or Figures as an evidence.

Tables III, IV, and V and the accompanied text  as mentioned as additional experimental results do not directly add more information. They can best be taken to an Appendix.

Please revise the first sentence in the second paragraph of Section VI.A.

It might be better to first refer to Fig. 3 in the results section. Also, Fig. 3 is too big with too many subfigures and information. You may split to a summarized concise Fig to be used in the main paper, and an extended detailed version to be taken to an Appendix.

Please revise the caption of Fig. 3.

**Strengths:**

The paper addresses a crucial research question with direct medical application: improving health disparity specially those related to racial and ethnic issues. The paper has studied whether the machine learning model make a different decision if the individual has a different sensitive attribute such as race.

**Summary Of The Paper:**

This paper evaluates how biased is the decision making in a clinical setting through the use of AI, namely, Counterfactual Fairness Regularization (CFReg), a casual and model-free algorithm with the goal of realizing a good trade-off between model fairness and supervised classification performance.

**Weaknesses:**

Some extended details in Section III Methods such as theories might be taken to an Appendix section.

From the 5 different dataset used, only 1 is related to health care. It would be more appreciated if all 5 dataset were related to health care, each of which related to a certain medical scenario such diabetic population, hypertensive, adults vs children, etc.

Maybe add a representative graph to compare the performance of the proposed algorithm (CF, CF-S) on 5 different data types.

---

### Official Review · Reviewer_SQQ5 · 2024-08-10
**Intuitive and interesting idea to evaluate and combat fairness in clinical decision making models**

**Overall Rating:** 8
**Confidence:** 3

**Other Quality Metrics:**

(a) Clarity of writing: Excellent
(b) Clinical Significance: Excellent
(c) Methodological Novelty: Excellent
(d) Experiments and Results: Excellent

**Questions For The Authors:**

Is this model only designed for tabular data that's most typical of EHR databases, or is this method potential extendable to other forms of data, just as text or images?

**Strengths:**

1. Very interesting and intuitive idea for evaluating fairness
2. The authors clearly laid out their algorithm pipeline and experimentation procedures
3. Potentially very wide use case, avoiding the expensive computation needed for full causal models
4. Introduction of a tunable parameter for the model to focus on model performance or fairness metric
4. Very clear and understandable writing

**Summary Of The Paper:**

The authors proposed a new modeling approach to account for fairness in clinical decision making models. The model is named CFReg, which utilizes CycleGAN and the idea of counterfactuals to learn mappings from different sensitive attributes to (counterfactual) features. The authors presented the algorithms as well as some proofs, and evaluated this model on 5 datasets, real-world and synthetic, on performance metrics as well as fairness scores.

**Weaknesses:**

1. Lack of a limitations sections, and lack of discussion on the generalizability of this method
2. Lack of application on a real EHR dataset, such as MIMIC III

---

### Official Review · Reviewer_xdDy · 2024-08-14
**A promising fairness approach for healthcare AI; needs clearer justification and comparisons.**

**Overall Rating:** 6
**Confidence:** 5

**Other Quality Metrics:**

(a) Clarity of writing: Good
Paper is generally well-written, but some sections(especialy the technical descriptions of CycleGAN adaptation and the theoretical analysis) need clearer explanations.

(b) Clinical Significance: Great
The work addresses the important issue of health disparities in AI-based clinical decision support systems

(c) Methodological Novelty: Good
The proposed causal model-free approach to counterfactual fairness, especially the use of CycleGAN for learning counterfactual mappings, is highly innovative.

(d) Experiments and Results: Good
The evaluation is comprehensive, covering multiple datasets including a healthcare dataset. However, comparison with recent state-of-the-art methods would further strengthen the results.

**Questions For The Authors:**

* How sensitive is the method to the quality of the learned mappings from CycleGAN? Poor mappings could lead to ineffective regularization.
* How does the method perform when the relationship between sensitive attributes and outcomes is more complex than in the tested datasets?
* How would the framework be extended to multi-category or continuous sensitive attributes,
* What is the exact procedure for incorporating the counterfactual regularization term into different types of machine learning models used in clinical decision support?
* what is the process for selecting the optimal λ parameter in practice for healthcare applications, balancing fairness and predictive performance?

**Strengths:**

* The paper proposes a novel causal model-free approach to counterfactual fairness, which is particularly relevant for complex healthcare datasets where causal relationships may be unclear or difficult to model.
* The theoretical analysis in Section IV provides a solid foundation for the practical results, particularly in relating counterfactual fairness to demographic parity.
* The paper clearly presents the trade-offs between fairness and accuracy, which is crucial for real-world applications.
* The use of CycleGAN for learning counterfactual mappings is an innovative approach to the problem of counterfactual fairness.

**Summary Of The Paper:**

The paper presents CFReg, a causal model-free algorithm for ensuring counterfactual fairness in machine learning systems, particularly for clinical decision support applications. The method uses CycleGANs to learn counterfactual mappings and incorporates these into a regularization term for training fair classifiers. The authors propose a novel regularization technique and evaluation metric, and validate their approach on multiple datasets, including a healthcare dataset and four benchmark datasets related to social fairness.

**Weaknesses:**

* The description of the healthcare dataset (Section V.A.2) is limited, making it difficult to assess the relevance of the results to real-world clinical decision support systems.
* The justification for why a causal model-free approach is necessary or superior in healthcare contexts is inadequate, given the importance of causal understanding in medicine.
* The paper lacks comparison with recent state-of-the-art fairness methods, particularly those designed for healthcare applications. This is a significant weakness as it makes it difficult to assess the relative performance and advantages of CFReg. Some recent state-of-the-art methods that could have been considered for comparison include: Fair Adversarial Gradient Tree Boosting (FAGTB) by Vincent Grari et.al.,Causal Multi-Level Fairness by Vishwali et.al. , etc.

* Minor comments:

* Figure 2: The diagram of the CycleGAN approach could benefit from additional labeling. For instance, the meaning of Gf and Gcf should be explained in the caption.
* Tables I-V: Consider adding standard deviations to the reported metrics. This would provide a more complete statistical picture of the results.
* The paper would benefit from a glossary of fairness-related terms (e.g., demographic parity, counterfactual fairness) to make it more accessible to a broader audience.

---

### Official Review · Reviewer_CtRi · 2024-08-16
**Fairness models to reduce disparity in machine learning models**

**Overall Rating:** 7
**Confidence:** 4

**Other Quality Metrics:**

(a) Clarity of writing: great
(b) Clinical Significance: great
(c) Methodological Novelty: good
(d) Experiments and Results: good

**Questions For The Authors:**

- The manuscript would benefit from describing the dataset and how balanced the datasets used in the study are regarding the sensitive attribute in each of them?
- Explain how the accuracy is not compromised in the CF models in details.
-Why the title of the  paper is is ".. mitigating effect of health disparity" while I dont think the paper is about health disparity just one case out of four is a healthcare dataset. If you think otherwise, please explain what you mean by health disparity.

**Strengths:**

- Through clear writing and a logical structure, the paper effectively communicates the nuances of fairness and counterfactual fairness, guiding the reader seamlessly through the presented concepts and models.

-The issue of fairness in healthcare is both crucial and delicate, requiring nuanced approaches and ethical considerations.

- Having such model that is computationally less expensive and can have both performance and fairness is huge effort.

**Summary Of The Paper:**

The quality of data is paramount in machine learning.  Recognizing that biases in datasets can amplify unfairness, the authors present an algorithm designed to promote causal fairness in supervised models and tested it in different datasets including healthcare, school admission, income and crime prediction.

**Weaknesses:**

- It would be good to briefly explain other efforts on fairness in machine learning especially in healthcare and elaborate on what is the advantage of the proposed method in the manuscript.

---

### Decision · Program_Chairs · 2024-09-23

Accept